# A blueprint for a synthetic genetic feedback optimizer

Andras Gyorgy [1] ✉, Amor Menezes [2] & Murat Arcak[3]

Biomolecular control enables leveraging cells as biomanufacturing factories. Despite recent advancements, we currently lack genetically encoded modules that can be deployed to dynamically fine-tune and optimize cellular performance. Here, we address this shortcoming by presenting the blueprint of a genetic feedback module to optimize a broadly defined performance metric by adjusting the production and decay rate of a (set of) regulator species. We demonstrate that the optimizer can be implemented by combining available synthetic biology parts and components, and that it can be readily integrated with existing pathways and genetically encoded biosensors to ensure its successful deployment in a variety of settings. We further illustrate that the optimizer successfully locates and tracks the optimum in diverse contexts when relying on mass action kinetics-based dynamics and parameter values typical in *Escherichia coli*.

Biological systems continuously sense and respond to changes in their environment by relying on sophisticated regulatory mechanisms to ensure both optimality and robustness, eventually leading to considerable complexity[1]. Synthetic biology seeks to (re)program cellular processes by designing and combining standardized biological parts in a modular fashion[2], often enhanced by computational approaches[3], to deploy synthetic gene circuits in a variety of contexts that range from biotherapeutics to environmental sciences[4].

Unfortunately, genetic modules regularly fail and require multiple design iterations[5] as a result of unmodeled interactions among circuit components and the host organism[6,7], frequently resulting in perplexing behaviors[8,9]. Further exacerbating this problem, genetic modules developed in one strain can behave fundamentally differently when deployed in other organism chassis[10,11]. Therefore, context-dependence presents a major obstacle to rationally and robustly controlling cellular processes, to creating novel functionalities, and to fine-tuning existing ones[6,12]. Metabolic burden, for instance, couples the expression of independent proteins and can significantly reduce cellular growth rate[13-19] (Fig. 1a).

While screening for optimal genetic realizations is now well-established by varying a diverse array of biochemical properties[20-23], such static approaches are unable to adaptively respond to

disturbances and shifts in the environment. To address this challenge, dynamic gene expression control[24-26] offers a promising solution, using a system-level approach that combines quantitative tools from a wide array of disciplines[27-30]. Recently, an especially fruitful direction has been to borrow ideas from control theory to analyze and design genetic feedback systems[31,32]. Notably, integral feedback has been proposed[33] to ensure perfect adaptation, that is, to return to a desired setpoint after a perturbation. Although it is not without limitations[34], this simple biological feedback control module has been successfully implemented in a variety of settings: in vivo[35,36] using both RNA-based and protein-based mechanisms, and in vitro[37,38] in bacteria, as well as in mammalian cells[39], and the design has even been extended in the form of a proportional-integral-derivative controller[40].

While these results represent a major leap in our ability to robustly control cellular behavior, they rely on an explicitly defined reference signal. Often times, however, the desired reference signal is defined only implicitly (i.e., where performance is maximized), and it continuously shifts due to fluctuations in the cellular environment, for instance, to ensure optimal resource re-allocation for maximal growth[41]. Therefore, we must develop genetic modules that can locate and track these implicit reference signals to create synthetic circuits

[1]Division of Engineering, New York University Abu Dhabi, Abu Dhabi, UAE. [2]Department of Mechanical and Aerospace Engineering, University of Florida, Gainesville, FL, USA. [3]Department of Electrical Engineering and Computer Sciences, University of California, Berkeley, CA, USA.
✉e-mail: andras.gyorgy@nyu.edu

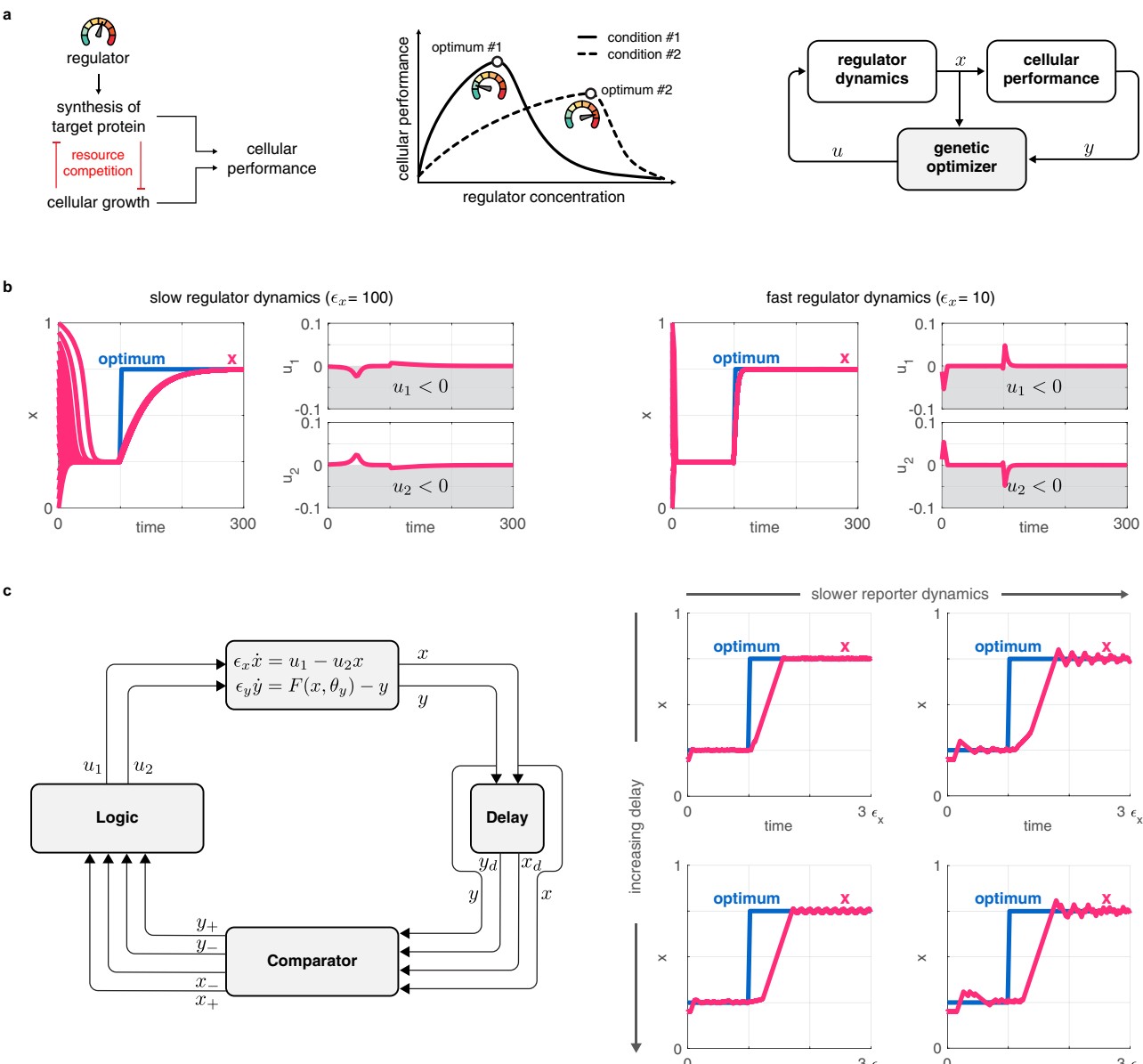

**Fig. 1 | Genetic optimizers can ensure maximal cellular performance.** Simulation parameters and further details are provided in Supplementary Section 5. **a** Population-level production of a target gene is maximized when growth rate and cellular synthesis rate are balanced[93]. The corresponding optimal concentration of a regulator may depend on both cellular and environmental conditions, and can be automatically adjusted by a genetic optimizer. **b** Gradient-based optimization can successfully track the time-varying optimum, but cannot be immediately translated to a genetic circuit because it may result in infeasible negative quantities. Decreasing $\epsilon_x$ yields faster convergence at the expense of greater control inputs $u_1$ and $u_2$. **c** Calculating $u_1$ and $u_2$ based on the trend of $x$ and $y$ ensures convergence to the optimum $x^*$. In the four panels at right, $\epsilon_y$ increases by an order of magnitude going from left to right (leading to slower $y$ dynamics), and the delay $t_d$ increases by an order of magnitude going from top to bottom.

that can sense and respond to environmental changes and adjust their activity for optimal performance[25,26].

To address this central challenge, here we develop and present a blueprint for a genetic optimizer module that dynamically adjusts the production and decay rates of a set of regulator species to ensure optimal cellular performance. Our approach relies on interconnecting common biomolecular sensors and actuators (e.g., logic gates, bistable switches, oscillators) that are readily available, using the traditional feedback loop architecture (Fig. 1a). As we illustrate through multiple application examples, our optimizer successfully locates and tracks an implicitly defined and time-varying optimum in diverse contexts when relying on mass action kinetics-based dynamics and parameter values typical in *Escherichia coli* (*E. coli*). While the basic idea underpinning our results can be most easily understood considering idealized

modules (e.g., completely symmetric toggle switches, absence of dilution), we also probe these assumptions and show that our approach works well even when considering non-ideal implementations. We further demonstrate that closed loop performance is robust to stochastic noise and disturbances (e.g., time-varying parameters due to various sources of context-dependence), and that our proposed optimizer can be easily integrated with existing pathways and genetically encoded biosensors to ensure its successful deployment in a variety of settings.

## Results
### Problem formulation
Our goal is to construct a genetic network to control the concentration of a set of regulator species $x$ so that cellular performance $F(x, \theta_y)$ is

maximized, where $\theta_y$ are (possibly time-dependent) biophysical parameters. We only assume that this performance is measurable via the reporter signal $y$, but it is otherwise unknown. We take the optimal concentration $x^*$ as a reference signal, and we seek to implement reference tracking where the time-varying setpoint is defined only implicitly (i.e., where performance is maximized). Finally, we wish to achieve this tracking (i) robustly, that is, in the presence of disturbances, uncertainties, and noise; and (ii) by relying on standard biological parts that are readily available and can be easily combined.

Let $x \in \mathbb{R}^n$ and $y \in \mathbb{R}$ denote the concentration vector of the controlled regulator species and the reporter (proxy for cellular performance) that we wish to maximize by adjusting the control signal $u \in \mathbb{R}^k$. The dynamics of the regulator species take the general form of

$$\epsilon_x \dot{x} = f(x, u, \theta_x), \qquad (1)$$

where dot denotes differentiation with respect to time, $\theta_x$ is a vector of (possibly time-dependent) parameters (e.g., transcription and translation rates, dissociation constants, degradation rates), and $\epsilon_x$ characterizes the timescale on which $x$ evolves. Regarding the reporter, we consider the general formulation of

$$\epsilon_y \dot{y} = F(x, \theta_y) - y \qquad (2)$$

to account for potential regulatory delays, where $\theta_y$ contains time-dependent parameters as above, and $\epsilon_y$ characterizes the timescale on which $y$ evolves. The lag between $x$ and $y$ increases with $\epsilon_y/\epsilon_x$: for instance, $\epsilon_y/\epsilon_x \to 0$ corresponds to the case when changes in $x$ are observable in $y$ instantaneously according to $y = F(x, \theta_y)$.

Considering the dynamics (1)–(2), we seek a feedback optimizer (Fig. 1a) of the form

$$\epsilon_z \dot{z} = g_z(z, x, y, \theta_z), \qquad u = g_u(z, \theta_u), \qquad (3)$$

where $z$ and $u$ are the internal state and the output of the optimizer, respectively, together with their time-dependent parameter vectors $\theta_z$ and $\theta_u$, and $\epsilon_z$ characterizes the timescale of the optimizer species. Although any one of $\epsilon_x$, $\epsilon_y$, and $\epsilon_z$ can be eliminated by rescaling time (Supplementary Section 1.1), we keep them all to illustrate their effects throughout the paper.

Assuming that $F(x, \theta_y)$ has a unique global time-dependent optimum at $x^*$ (if this is not the case, our approach only yields local optimality), we seek to design the optimizer to ensure that (i) trajectories of the closed loop system (1)–(3) converge towards the unique optimum $x^*$; and (ii) the integrated system is biologically realizable and relies only on non-negative signals.

### Gradient ascent and approximate gradient ascent

To illustrate the main idea underpinning the proposed optimizer module, we first consider a single species $x$ that is produced and degraded according to $\epsilon_x \dot{x} = u_1 - u_2 x$ (Supplementary Section 1.1), where $u_1$ and $u_2$ represent control signals regulating the production and degradation of $x$, respectively. In the absence of biological constraints, a potential choice for the optimizer is a gradient-based system with the feedback law $u_1 = \nabla_x F(x, \theta_y)$ and $u_2 = -\nabla_x F(x, \theta_y)$, where $\nabla_x F(x, \theta_y)$ denotes the gradient of $F(x, \theta_y)$. The closed loop dynamics $\epsilon_x \dot{x} = (1 + x)\nabla_x F(x, \theta_y)$ realizes gradient ascent by converging to $x = x^*$ where $\nabla_x F(x, \theta_y) = 0$, thus tracking the time-varying optimum (Fig. 1b). Moreover, it is sufficient to simply swap the signs in the control laws above to implement minimization (gradient descent). Unfortunately, this approach has two fundamental limitations: (i) it requires explicit knowledge of the gradient $\nabla_x F(x, \theta_y)$; and (ii) $u_1$ and $u_2$ can become negative (Fig. 1b, gray regions).

The gradient-based control law above can be phrased as: $x$ should be increased if the gradient of cellular performance is positive, and

decreased otherwise. To overcome the first issue (reliance on the gradient), we can rephrase the control law as: $x$ should be increased if $x$ and $y$ are increasing or decreasing together, and decreased otherwise. To implement this, we must keep track of the change in both $x$ and $y$: let $w_d(t) = w(t - t_d)$ for the time delay $t_d > 0$ so that $\Delta x = x - x_d$ and $\Delta y = y - y_d$ represent the changes in $x$ and $y$, respectively. With this, $x$ should be increased when $\Delta x \Delta y > 0$, and decreased otherwise (Fig. 1a), corresponding to $u_1 > 0$, $u_2 = 0$ and $u_1 = 0$, $u_2 > 0$, respectively. Unfortunately, this solution still suffers from the second issue highlighted above, as $\Delta x$ and $\Delta y$ can take on negative values. To avoid this, we must keep track of whether $x$ is increasing or decreasing via the non-negative indicator signals $x_+ = h(x - x_d)$ and $x_- = h(x_d - x)$ where $h(\cdot)$ is defined as $h(w) = 1$ for $w > 0$ and $h(w) = 0$ otherwise. The indicator signals $y_+, y_- \geq 0$ are defined similarly. With this, $x$ needs to be increased ($u_1 > 0$, $u_2 = 0$) when either $x_+, y_+ > 0$ or $x_-, y_- > 0$, and it needs to be decreased ($u_1 = 0$, $u_2 > 0$) otherwise. This can be implemented via the control law

$$u_1 = x_+ y_+ + x_- y_-, \qquad u_2 = x_+ y_- + x_- y_+ \qquad (4)$$

to ensure that trajectories of the closed loop system converge towards the optimum (Fig. 1c), largely unaffected by the value of $\epsilon_y$ and $t_d$ (Supplementary Fig. 1). As expected, however, once the lag between $x$ and $y$ (governed by $\epsilon_y$), or the delay between $x$ and $x_d$ or $y$ and $y_d$ (governed by $t_d$) approach the timescale of the regulator dynamics (governed by $\epsilon_x$), closed loop performance quickly deteriorates.

In summary, the proposed optimizer overcomes the major issues of the gradient-based solution (reliance on the gradient and negative signals) without compromising closed loop performance. As illustrated in Fig. 1c, this is achieved by combining three crucial modules: (i) delay of $x$ and $y$ to obtain $x_d$ and $y_d$, respectively; (ii) comparison of $x$ with $x_d$ to obtain the indicator signals $x_+$ and $x_-$, and similarly for $y$; and (iii) computation of the control signals $u_1$ and $u_2$ based on the indicators according to (4). Before presenting the whole integrated system, we next detail how these idealized and abstract functions can be realized individually by relying on standard synthetic biology modules, and how their biophysical parameters should be selected for optimal performance that is robust to stochastic noise and disturbances.

### Delay module

The delayed signal $x_d$ that tracks $x$ with a lag can be implemented using a variety of modules that selectively synthesize/activate the former only in the presence of the latter (and similarly for $y_d$ and $y$), as illustrated in Fig. 2. Such a positive relationship can be captured via the dynamics

$$\epsilon_d \dot{x}_d = \alpha_d x - x_d, \qquad \epsilon_d \dot{y}_d = \alpha_d y - y_d, \qquad (5)$$

where $\alpha_d \approx 1$ denotes the production rate constant (following non-dimensionalization, see Supplementary Section 1.2), whereas $\epsilon_d$ regulates the timescale of the delay module (e.g., greater $\epsilon_d$ yields slower dynamics). When $\alpha_d = 1$, we obtain that $x_d \to x$ and $y_d \to y$. Thus, the signals $x_d$ and $y_d$ track $x$ and $y$ without steady state error (in case of constant reference signals). We first assume that the delay module is tuned to guarantee perfect tracking ($\alpha_d = 1$). Later, we will demonstrate that the optimizer works even when this is not ensured (provided that $\alpha_d \approx 1$).

The value of $\epsilon_d$ needs to be sufficiently small so that tracking is fast enough and the optimizer remains responsive to changes in $x$ (and $y$). This observation from Fig. 1c is echoed in the simulation data presented in Fig. 2: as long as the timescale on which the reporter and the delayed signals evolve is sufficiently fast compared to the dynamics of the regulator (i.e., $\epsilon_d, \epsilon_y \ll \epsilon_x$), closed loop performance of the optimizer module is largely unaffected by the exact value of $\epsilon_d$ and $\epsilon_y$. Unsurprisingly, there is a sharp decline in closed loop performance as the timescale of either the reporter or the delay module approaches that of the regulator (Supplementary Fig. 2).

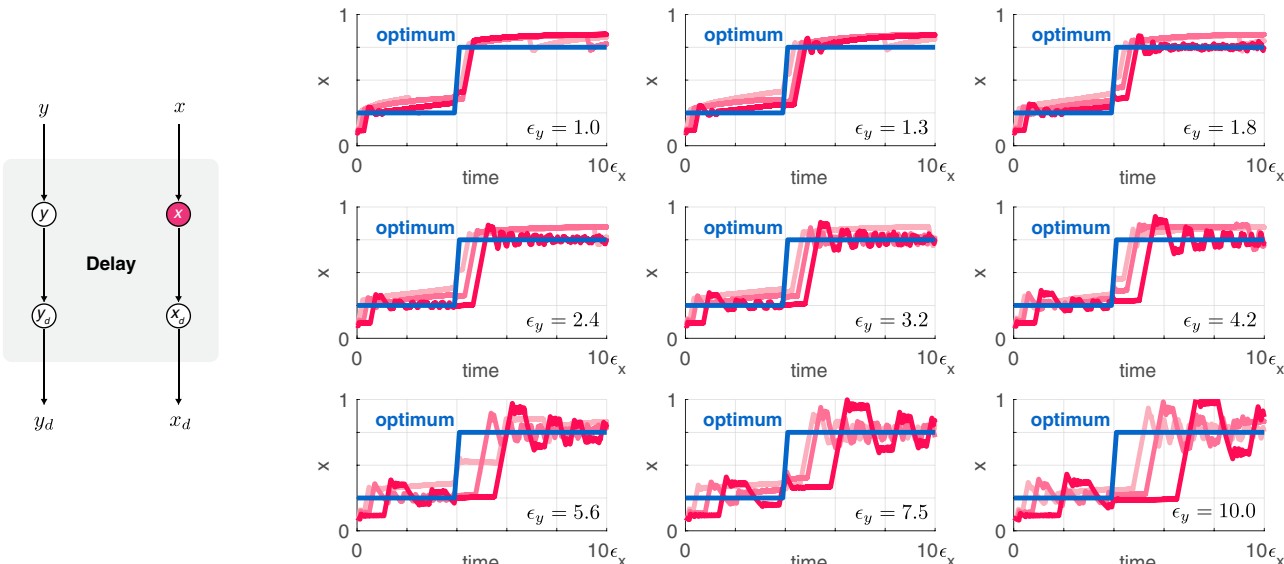

**Fig. 2 | The delay module ensures tracking of the regulator and the reporter signals.** Light, medium, and dark red correspond to $\epsilon_d = \epsilon_y/2$, $\epsilon_d = \epsilon_y$, and $\epsilon_d = 2\epsilon_y$, respectively. The panel in the top left corner corresponds to $\epsilon_y = \epsilon_x/100$, and $\epsilon_y$ increases towards the lower right panel where $\epsilon_y = \epsilon_x/10$ (sample points are spaced equidistantly on a logarithmic scale). Simulation parameters and further details are provided in Supplementary Section 5.

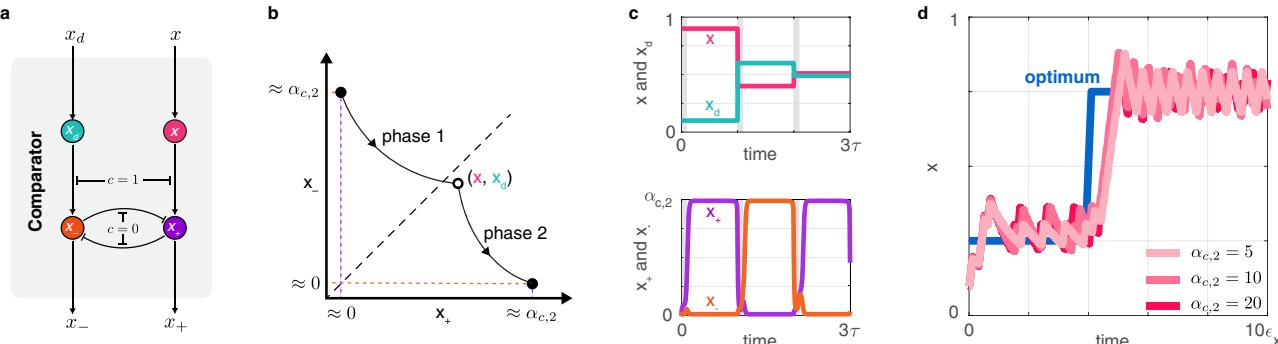

**Fig. 3 | The comparator module generates the indicator signals based on the actual and delayed signals for both the regulator and the reporter.** Simulation parameters and further details are provided in Supplementary Section 5. **a** The signal $c$ alternates between two states ($c = 0$ and $c = 1$) with period $\tau$, activating two different sets of regulatory interactions[94]. **b** During phase 1, $(x_+, x_-)$ tracks the reference $(x, x_d)$, whereas during phase 2, $(x_+, x_-)$ converges to either of the stable fixed points based on the sign of $x - x_d$. **c** The signals $x_+$ and $x_-$ switch between their ON and OFF states depending on whether $x < x_d$ or $x > x_d$ (phase 1 is depicted in gray). **d** Closed loop performance is largely unaffected by the value of $\alpha_{c,2}$.

## Comparator module

We next need to compare $x$ with $x_d$ and $y$ with $y_d$ to obtain the indicator signals $x_+$, $x_-$, $y_+$, and $y_-$ (Fig. 1c). The comparator module provides these signals by combining three parts: the delay module from the previous section, bistable switches[42], and an oscillator[43,44] to periodically activate/deactivate regulatory interactions (Fig. 3a).

For the sake of simplicity, here we assume that the output $c$ of a genetic oscillator alternates between two values: $c = 0$ and $c = 1$ during phase 1 and phase 2, respectively. We rely on this periodic signal to switch the comparator with inputs $x_d$ and $x$ and outputs $x_-$ and $x_+$ between two modes according to

$$\epsilon_c \dot{x}_- = (1-c)(\alpha_{c,1} x_d - x_-) + c\left(\frac{\alpha_{c,2}}{1+x_+^2} - x_-\right),$$

$$\epsilon_c \dot{x}_+ = (1-c)(\alpha_{c,1} x - x_+) + c\left(\frac{\alpha_{c,2}}{1+x_-^2} - x_+\right),$$

where $\alpha_{c,1}$ and $\alpha_{c,2}$ are production rate constants (de-dimensionalized, see Supplementary Section 1.3), and $\epsilon_c$ characterizes the timescale of the comparator dynamics.

During phase 1 ($c = 0$), the system behaves as the delay module in Fig. 2: the indicator signals $x_+$ and $x_-$ track the reference signals $x$ and $x_d$, respectively (Fig. 3b). Assuming that this tracking occurs on a timescale faster than that of $x$ and $x_d$, during phase 1 $x$ and $x_d$ are effectively frozen and $x_+$ and $x_-$ converge to these values if $\alpha_{c,1} = 1$ (Fig. 3c). Therefore, at the end of phase 1, the values of $x$ and $x_d$ are stored in $x_+$ and $x_-$, respectively.

During phase 2 ($c = 1$), the comparator behaves as a bistable switch to implement a memory module, such that the initial conditions of $x_+$ and $x_-$ are the frozen values of $x$ and $x_d$, respectively. A balanced toggle switch (i.e., $\alpha_{c,2}$ is identical for $x_+$ and $x_-$) forces convergence to the $x_+$-dominated stable fixed point if $x_+ > x_-$ at the beginning of phase 2 (i.e., if $x > x_d$ at the beginning of phase 1), and to the $x_-$-dominated stable fixed point if $x_- > x_+$ at the beginning of phase 2 (i.e., if $x_d > x$ at the beginning of phase 1). Importantly, the location of these stable fixed points is independent of the difference between $x$ and $x_d$ at the beginning of phase 1 (Fig. 3b and Supplementary Section 1.3). Therefore, at the end of phase 2, the output of the comparator depends only on the sign of $x - x_d$ (at the beginning of phase 1), but not on its magnitude (Fig. 3c), a crucial feature of the proposed design to ensure the compatibility of dynamic ranges between the comparator and the logic module (Fig. 1c). Another comparator with inputs $y_d$ and $y$ and outputs $y_-$ and $y_+$ behaves similarly.

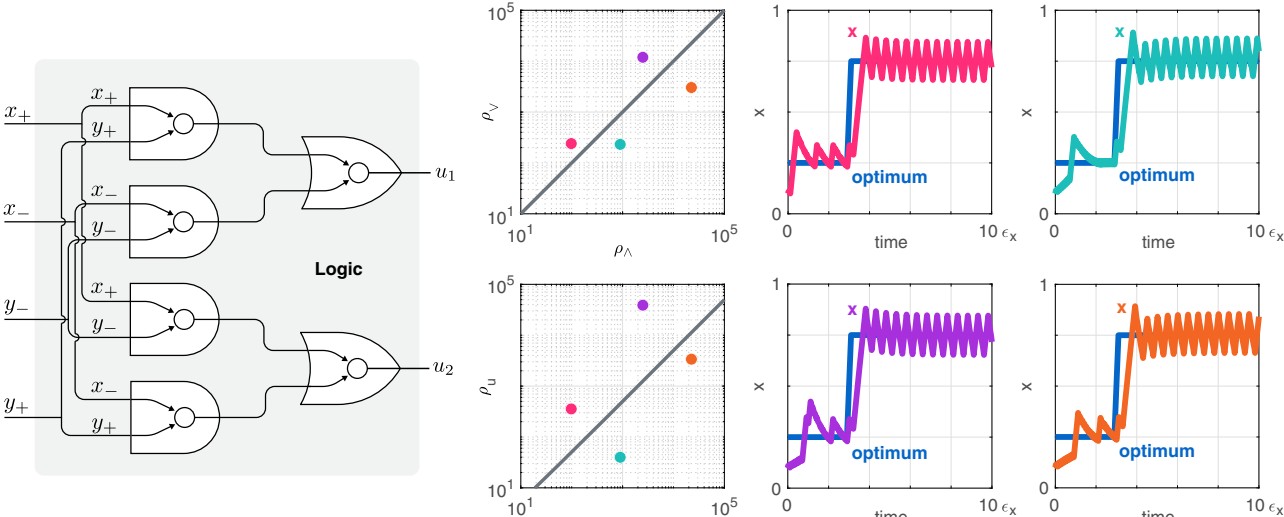

**Fig. 4 | The logic module combines the indicator signals to generate the control signals.** The dynamic range of the signals in the input, middle (between the AND and OR gates), and output layer of the logic module is denoted by $\rho_\wedge$, $\rho_\vee$, and $\rho_u$, respectively. While selecting the dissociation constants $K_\wedge$ and $K_\vee$ in the geometric mean of the respective dynamic ranges may be an intuitive choice (yielding the dark gray lines), the performance of the optimizer displays significant robustness to deviations from this particular baseline choice when tracking the optimum value $x^*$ (blue). Colored circles correspond to different choices of $\alpha_{c,2}$ (affecting the input dynamic range $\rho_\wedge$), together with substantial perturbations in the dissociation constants compared to the above specified baseline choice. Simulation parameters and further details are provided in Supplementary Section 5.

While $\alpha_{c,1} \approx 1$ is essential for precise tracking, the behavior of the optimizer is considerably less impacted by the value of $\alpha_{c,2}$ as long as bistability is ensured ($\alpha_{c,2} > 2$). For instance, closed loop performance and the time it takes to reach either of the stable fixed points display negligible dependence on $\alpha_{c,2}$ (Fig. 3d and Supplementary Fig. 3b). Although here we assume that the bistable switch is perfectly balanced, we later demonstrate that closed loop performance does not appreciably deteriorate in the presence of moderate levels of imbalance.

## Logic module

The final module in the optimizer (Fig. 1c) combines the indicator signals $x_+$, $x_-$, $y_+$, and $y_-$ to generate $u_1$ and $u_2$ by approximating the control law in (4). This can be done by relying on standard logic gates[45,46], as illustrated in Fig. 4. For the sake of simplicity, here we assume that these logic gates operate on a faster timescale than any other module in Fig. 1c (Supplementary Section 1.4). Therefore, the behavior of the logic gates can be approximated by utilizing their quasi-steady state input-output mappings[45]. In particular, considering the common mathematical model of AND ($\wedge$) and OR ($\vee$) gates with input signals $A$ and $B$ (Supplementary Section 1.4), the outputs are given by

$$H_\wedge(A,B) = \frac{A^n}{A^n + K_\wedge^n}\frac{B^n}{B^n + K_\wedge^n}, \qquad H_\vee(A,B) = \frac{A^n}{A^n + K_\vee^n} + \frac{B^n}{B^n + K_\vee^n},$$

with Hill coefficient $n$ and dissociation constants $K_\wedge$ and $K_\vee$ for the AND and OR gates, respectively. The input-output mapping of the proposed circuit in Fig. 4 is thus given by $u_1 = H_\vee\left(H_\wedge\left(x_+,y_+\right), H_\wedge\left(x_-,y_-\right)\right)$ and $u_2 = H_\vee\left(H_\wedge\left(x_+,y_-\right), H_\wedge\left(x_-,y_+\right)\right)$.

We next investigate how the dynamic range of the indicator signals, the Hill coefficient, and the dissociation constants of the logic gates affect closed loop performance. As illustrated in Fig. 4, the behavior is insensitive to the dissociation constants $K_\wedge$ and $K_\vee$, to the input dynamic range (determined by $\alpha_{c,2}$, see Supplementary Section 1.3), and to the Hill coefficient $n$ (Supplementary Fig. 4). Thus, the value of these parameters represent non-critical design choices, unlike the timescale on which the logic module evolves. As expected, the optimizer successfully locates the optimum as long as the dynamics of the logic module evolve on a timescale faster than that of the comparator module (so that the quasi-steady state approximation above is accurate on the slower timescale of the system), however, further speed reduction causes closed loop performance to quickly deteriorate (Supplementary Fig. 5).

## Closed loop performance of the simplified optimizer

Before outlining a concrete molecular implementation of the whole integrated system, here we analyze closed loop performance and accuracy when considering the abstract modules introduced above. We first assume that the delay module ensures reference tracking with no error and that the comparator module realizes perfect comparison of its input signals when generating the output indicator signals. We then focus on a less ideal but more practical realization of the optimizer module, and reveal how the presence of errors in delay and comparison affect closed loop performance.

The time it takes to approach the optimum $x^*$ depends on the position of the initial value $x_0$ relative to $x^*$. However, closed loop performance after this initial transient is virtually identical for all $(x_0, x^*)$ pairs in the absence of tracking and comparison error (Supplementary Fig. 6a). Importantly, closed loop performance is robust to environmental disturbances that cause drastic shifts in the optimum and also to noise (Fig. 5a). Furthermore, as long as the delay module evolves on a timescale faster than the regulator ($\epsilon_d \ll \epsilon_x$), closed loop performance is insensitive to the value of $\epsilon_d$, as expected: e.g., a 10-fold increase in the value of $\epsilon_d$ yields virtually no change (red and green curves in Fig. 5a).

Once the optimum is reached, trajectories oscillate around it due to the periodic behavior of the comparator module. The amplitude of these oscillations increases with $\tau/\epsilon_x$, where $\tau$ is the period of the oscillator in the comparator module and $\epsilon_x$ characterizes the timescale of the regulator dynamics (Supplementary Section 1.5). This result can be interpreted as follows: lower values of $\epsilon_x$ correspond to faster control (yielding more rapid changes in $x$ over the same time interval), whereas greater values of $\tau$ correspond to longer durations when the control action is maintained (but not recalculated and adjusted even if the optimum is reached and crossed). Therefore, decreasing $\epsilon_x$ or

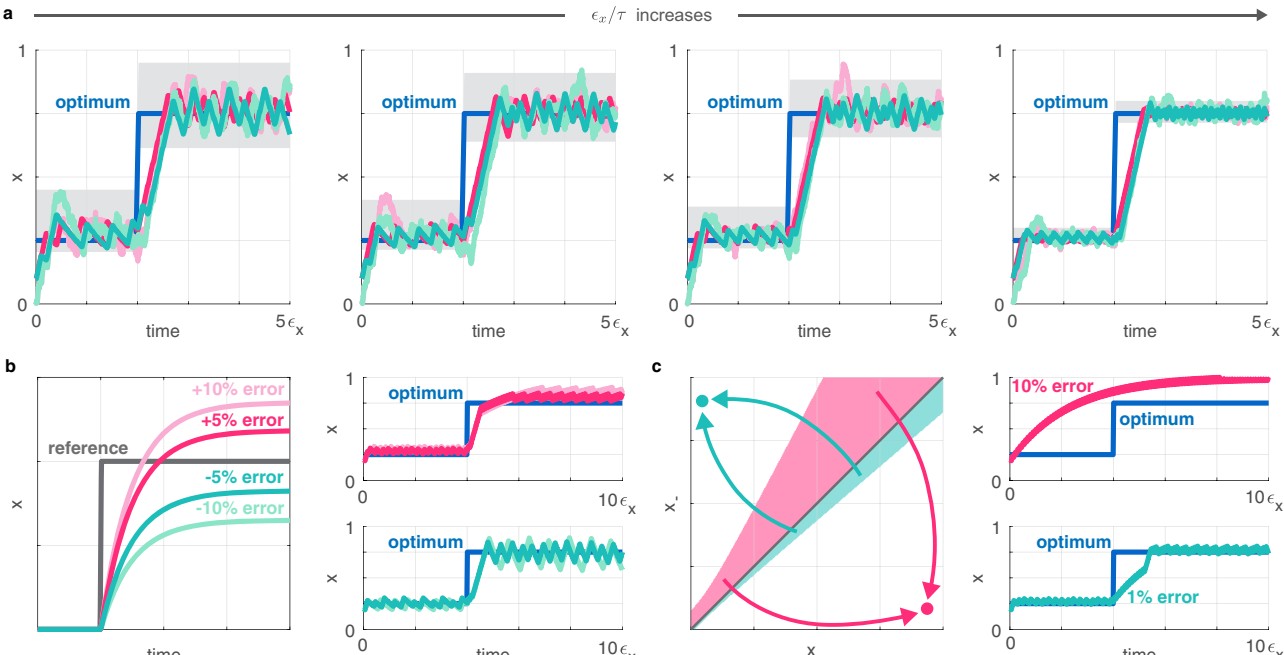

**Fig. 5 | Closed loop performance and accuracy with the simplified dynamics.**
Simulation parameters and further details are provided in Supplementary Section 5. **a** In the absence of additive noise (dark red and dark green), trajectories are confined within the gray region around the time-varying optimum $x^*$ (blue) when $\epsilon_y = 0$ (Supplementary Section 1.5). In the presence of stochastic noise (light red and

light green), closed loop trajectories may temporarily leave this region. The value of $\epsilon_d$ is 10-times greater for (light and dark) green than for (light and dark) red. **b** Performance decreases as the tracking error in the delay module increases. **c** Shaded regions correspond to initial conditions such that trajectories converge to an incorrect stable fixed point.

increasing $\tau$, thus increasing the ratio $\tau/\epsilon_x$, eventually leads to more pronounced (occasional) steps in the wrong direction, before these errors are inevitably corrected. These unwanted oscillations can be mitigated by rendering the dynamics of the regulator slower (increasing $\epsilon_x$), or by selecting a clock with shorter period (decreasing $\tau$).

The results so far are underpinned by two crucial assumptions: (i) perfect tracking by the delay module ($\alpha_d = 1$); and (ii) perfect comparison by the comparator module ($\alpha_{c,2}$ is identical for both species). Therefore, we next explore how closed loop performance is affected when these assumptions do not hold true. Tracking occurs with steady state error when $\alpha_d \neq 1$ (Fig. 5b), thus we may erroneously conclude that $x$ is increasing even though $x_d > x$, simply because $\alpha_d > 1$ (Fig. 5b). Similarly, unless $\alpha_{c,2}$ is identical for both species in the comparators, comparison of the indicator signals $x_-$ and $x_+$ can introduce error as trajectories may incorrectly converge to the $x_+$-dominated stable fixed point even when $x_- > x_+$ at the start of phase 2 (red in Fig. 5c). These errors then propagate through the logic module, eventually leading to incorrect control signals. Tracking error ($\alpha_d \neq 1$) leads to more pronounced oscillations around the optimum, whereas substantial asymmetry in $\alpha_{c,2}$ can render the optimizer unreliable (red in Fig. 5c). However, reducing this asymmetry to more modest levels (for instance, via RBS and promoter engineering, or by leveraging decoy sites, see Supplementary Section 1.3) eliminates such performance degradation (green in Fig. 5c). Therefore, while closed loop performance may deteriorate with overwhelming errors in either tracking or comparison, the optimizer displays robust behavior in the presence of moderate levels of error.

## Implementation with existing synthetic biology modules
The proposed genetic optimizer can be readily implemented by combining existing synthetic biology modules relying on standard parts and components, as we illustrate in Fig. 6. To ensure correct functioning of the optimizer, modules evolve on two different

timescales: the regulator, the reporter, the delay module, and the phase selector oscillator are governed by dynamics that are slower than those of the toggle switch-based memory modules and the logic gates, which guides the design choices in Fig. 6.

Production and removal of the regulator $x$ is coordinated by the outputs of the logic module: while $u_1$ activates synthesis of $x$, $u_2$ stimulates its degradation by upregulating the transcription of the protease $v$ that targets $x$ by recognizing the degradation tag fused to it. For instance, appending the *Mesoplasma florum* (*M. florum*) ssrA tag (mf-ssrA) to the C terminus of $x$ results in its inducible degradation upon expression of the *M. florum* Lon protease (mf-Lon) $v$, operating independently from and orthogonally to the host degradation pathways[47,48]. Alternatively, instead of expressing the mf-Lon protease, $v$ can encode the adapter SspB, tethering protein targets fused with a modified ssrA tag to ClpXP for controlled degradation[49]. To ensure that the removal of $x$ is primarily due to controlled degradation via $v$, the regulator is also equipped with weak self-activation to balance the impact of dilution due to cell growth.

The rest of the genetic optimizer operating on the slower timescale can be implemented using common transcription factors[50]. For instance, the phase selector oscillator can be realized using the repressilator[43,44] that consists of three repressor proteins, one of them ($r$) co-expressed with an activator ($a$), so that these two can together coordinate periodic shifting between phase 1 and phase 2 of the comparators. Similarly, transcriptional control underpins the implementation of delaying $x$ and $y$ to obtain $x_d$ and $y_d$. While in Fig. 6 we assume that the regulator $x$ impacts $y$ via transcriptional activation and that $y$ acts as an activator, crucially, the proposed optimizer module works even when the impact of $x$ on $y$ is mediated via more complex pathways, or when the reporter acts as a repressor, as we demonstrate in the subsequent section.

As RNA-based solutions enable considerably faster information processing, the indicator signals are implemented by relying on CRISPRi, thus $x_+$, $x_-$, $y_+$, and $y_-$ are gRNAs. During phase 1, $x_+$ and $x_-$ are

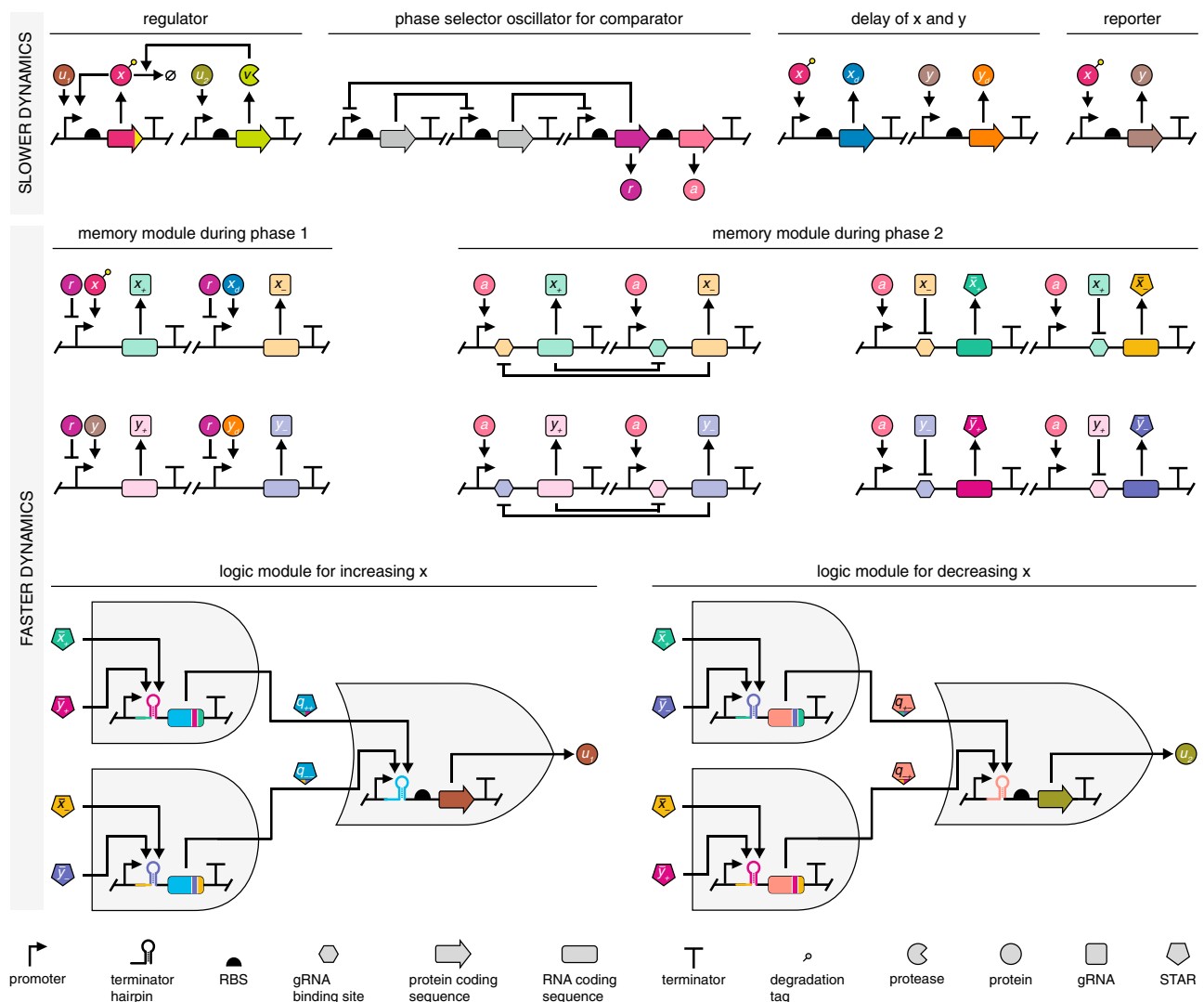

**Fig. 6 | Genetic layout of the optimizer module.** The realization relies on genetic parts and modules that are already available, in particular: (i) protein-based transcriptional control[50]; (ii) inducible degradation via the *M. florum* Lon protease and ssrA tag[47,48]; (iii) a repressilator-based oscillator[43,44]; (iv) CRISPRi-based toggle switches[51]; and (v) STAR-based logic gates[53,54]. The mass action kinetics-based mathematical model underpinning the dynamics of the integrated system is included in Supplementary Section 2.1, together with detailed discussion of the typical range of model parameters in Supplementary Section 2.2, and their selected values in Supplementary Table 1. Here, we assume that the host genome is already equipped with a dCas9 expression cassette[95], otherwise the optimizer must also include it.

expressed from promoters under the induction of $x$ and $x_d$ to track them with a delay. During phase 2, $x_+$ and $x_-$ mutually repress each other to form a CRISPRi-based bistable switch by recruiting dCas9[51]. Comparison of $y$ and $y_d$ is realized similarly via $y_+$ and $y_-$. To alternate between phase 1 and phase 2, promoters of the gRNAs are under the coordinated control of the repressor $r$ and the activator $a$ (Fig. 6). To ensure reliable switching between the phases, promoter leakiness could be minimized via stringent multi-level control of gene expression[52].

Finally, the logic module is realized using STAR-based AND and OR gates due to their highly programmable and orthogonal nature[53,54]. To interface these with the CRISPRi-based indicator signals, we first co-express STARs ($\bar{x}_+$, $\bar{x}_-$, $\bar{y}_+$, and $\bar{y}_-$) with the gRNAs of the comparator module ($x_+$, $x_-$, $y_+$, and $y_-$) during phase 2. This can be achieved by duplicating the transcriptional units encoding gRNAs, only this time expressing STARs (Fig. 6). As a result, the AND gates can take these STARs that serve as proxy for the indicator signals to generate the intermediate STARs $q_{++}$, $q_{+-}$, $q_{-+}$, and $q_{--}$, either by performing signal integration at the level of the target RNA[53], or at the level of the STAR

by splitting it into two halves corresponding to the linear and the terminator hairpin binding regions alongside with an interaction sequence to promote their assembly[54]. Similarly, the OR gates take these intermediate STARs to regulate the expression of the transcription factors $u_1$ and $u_2$ by interacting with the intrinsic terminator hairpins that prevent their transcription (either $q_{++}$ or $q_{--}$ for the former, and either $q_{+-}$ or $q_{-+}$ for the latter). Production of $x$ is then modulated by $u_1$, whereas its controlled removal by $u_2$ (via $v$), thus closing the feedback loop.

Each module featured in Fig. 6 has already been successfully implemented, and considering the mass action kinetics-based model of the whole integrated system (Supplementary Section 2.1), the optimizer locates and tracks the optimum when relying on parameter values typical in *E. coli* (Supplementary Section 2.2). Furthermore, closed loop performance is robust to considerable parameter variations, ranging from changes in the period of the phase selector oscillator through shifts in the production and degradation rates of the regulator to the shape of the application-specific objective function (Supplementary Figs. 8–14). Next, we demonstrate that the optimizer

ensures equally robust closed loop performance in a wide variety of contexts.

## The optimizer tracks the optimum in diverse contexts

In Fig. 6, we considered the case when the regulator $x$ directly activates the production of the reporter $y$. Here, we illustrate not only that closed loop performance remains similar when considering more complex regulatory schemes and different objective functions (detailed mathematical models are provided in Supplementary Section 3), but also that the optimizer requires only minor modifications to implement minimization instead of maximization, or even multi-species optimization (Fig. 7). As a concrete application example, in Fig. 8 we demonstrate that the genetic optimizer can be deployed for optimal growth rate regulation in the presence of cellular stress. In all examples featured in Figs. 7 and 8, we rely on the mass action kinetics-based model of the whole integrated system outlined in Fig. 6 without any changes to the typical biophysical parameter values featured in Supplementary Table 1.

Starting with the case of simple activation in Fig. 7a, while the production rate of $y$ initially increases with $x$, the overall relationship between the regulator and the reporter may become non-monotonic due to metabolic burden[9]. Considering either a non-smooth objective function or a smooth bell shaped curve (Supplementary Figs. 11–14), the optimizer successfully locates the optimum, demonstrating that the shape of the objective function has negligible impact on closed loop performance. Closed loop performance also remains practically identical when the impact of $x$ on $y$ is mediated via a regulatory cascade that may even contain self-regulation (Fig. 7a). The behavior is robust to changes in the parameter values of the regulated pathway as well. To illustrate this, consider first the regulatory cascade in Fig. 7b equipped with a negative feedback loop. The optimizer quickly locates the new optimum and reliably tracks it subsequent to abrupt changes in the dissociation constants $\kappa_x$, $\kappa_w$, and $\kappa_z$. The optimum is also rapidly reached when $x$ impacts the expression of $y$ via an incoherent type-1 feed-forward loop after substantial perturbations in the dissociation constant $\kappa_x$, the inducer concentration $w$, or the production rate constant $\beta_z$ (Fig. 7b).

To locate and track the minimum instead of the maximum, gradient descent can be realized by simply swapping $u_1$ and $u_2$ either in the output stage of the logic module so that the former promotes removal of $x$ (via the protease $v$) and the latter its synthesis (alternative #1 in Fig. 7c), or when activating the expression of $x$ and $v$, respectively (alternative #2 in Fig. 7c). This may prove helpful, for instance, when we wish to maximize the concentration of the repressor $\tilde{y}$, thus it may not be used to directly obtain the delayed signal $y_d$. While we could co-express an activator together with $\tilde{y}$ and rely on the latter to maximize the former, this may result in additional metabolic burden and issues related to compositional context[8]. Alternatively, we can minimize a target $y$ of the repressor $\tilde{y}$, thus indirectly maximizing $\tilde{y}$ without any negative impact on closed loop performance (Fig. 7c).

To demonstrate that the proposed optimizer can be deployed even when multiple regulators need to be tuned together to locate the optimum, consider the multi-species optimization problem depicted in Fig. 7d. Instead of seeking the optimum $x^* = (x_1^*, x_2^*, \ldots, x_N^*)$ in all coordinates simultaneously, we focus on each controlled variable $x_i$ separately. By decomposing the multi-dimensional optimization problem into a collection of $N$ one-dimensional problems, each of them takes the form of the previously considered scalar case. Thus, we can deploy a realization of the controller from Fig. 6 for the regulation of each $x_i$ (their independently controlled removal can be realized by complementing the *M. florum*-based solution in Fig. 6 with orthogonal protease:cleavage site pairs from *Potyvirus*[55]), activated one-by-one in a cyclical structure, for instance, by using a ring oscillator with $N$ species[56], as illustrated in Fig. 7e. Data in Fig. 7d confirm that closed loop performance does not deteriorate even

when multiple species need to be regulated to ensure optimal performance.

Finally, consider growth rate control in the presence of cellular stress (Fig. 8). Termed as the speedometer of growth rate[57], the alarmone (p)ppGpp is the primary regulator of both growth and RNA synthesis during exponential growth[58,59] via a strong inverse relationship[60,61]. To ensure optimal (p)ppGpp concentration, cells carefully balance the expression levels of RelA/SpoT Homolog (RSH) proteins to catalyze the synthesis and hydrolysis of (p)ppGpp[62]. Leveraging this, the feedforward controller developed in ref. 63 and outlined in Fig. 8a can be deployed to exogenously adjust growth rate via SpoTH. While expression of SpoTH has a positive impact on growth rate at moderate levels as a result of (p)ppGpp hydrolysis[64], its over-expression can result in considerable metabolic burden, thus giving rise to the non-monotonic relationship between SpoTH expression and growth rate observed in ref. 63 and illustrated in Fig. 8b. By placing the protein $y$ under the control of an rrn P1 promoter, we create a proxy signal for ribosome production and growth rate[65,66], and ensure that the optimizer indirectly maximizes growth rate via $y$ by leveraging that rRNA synthesis is the rate-limiting step in ribosome production[58], yielding a strong inverse relationship between (p)ppGpp concentration and both ribosome synthesis and growth[67]. Data presented in Fig. 8c, d demonstrate that (i) optimal regulation of SpoTH, and thus of (p)ppGpp expression can substantially mitigate the negative impact of cellular stress on growth rate; (ii) performance with the optimizer approaches the theoretical optimum that can be achieved by careful exogenous tuning of SpoTH expression; and (iii) closed loop behavior is robust to considerable stochastic noise and parameter variations (Supplementary Figs. 16–20).

Taken together, these results demonstrate that the optimizer can be successfully deployed in a variety of contexts when relying on (i) an implementation combining existing synthetic biology parts and components; (ii) mass action kinetics-based dynamics; and (iii) biologically plausible parameter values typical in *E. coli*.

## Discussion

Synthetic biology promises to revolutionize multiple sectors ranging from metabolic engineering[25] to sustainable biomanufacturing[68]. By rearranging existing regulatory linkages and introducing novel ones, experimental techniques complemented by computational tools[69,70] offer a particularly lucrative research direction to utilize cells as microscopic factories for the dynamic regulation of metabolic pathways. Accordingly, recent years have seen an explosion of studies focusing on controller design for genetic systems[31–40,71] to ensure that synthetic circuits function robustly in different host organisms and cellular contexts. Complementing these efforts, the optimizer module developed here acts as a feedback controller and steers the regulated genetic system towards the optimum, but crucially, without having explicit knowledge of either its location, the objective function itself, or the pathway that needs to be regulated.

Our results also highlight how existing regulatory solutions based on exogenous control can be enhanced by combining them with the proposed genetic optimizer without compromising performance or requiring any modification to already developed pathways. Both the regulator $x$ and the reporter $y$ can be implemented using widely available transcription factors[50], and synthesis of $y$ can be made responsive to a multitude of signals and analytes relying on the vast collection of protein-based and RNA-based biosensors[72–75]. For instance, transcription of $y$ can be modulated by expressing it from the cognate promoter of a ligand inducible transcription factor[73,76], whereas metabolite-sensing riboswitches can be deployed to directly impact translation initiation[77,78]. The optimizer can thus be easily interfaced with the vast collection of biosensors that is continuously expanded via genome mining, rational design, and screening based on directed evolution[74,75,79,80]. Consequently, we expect that our

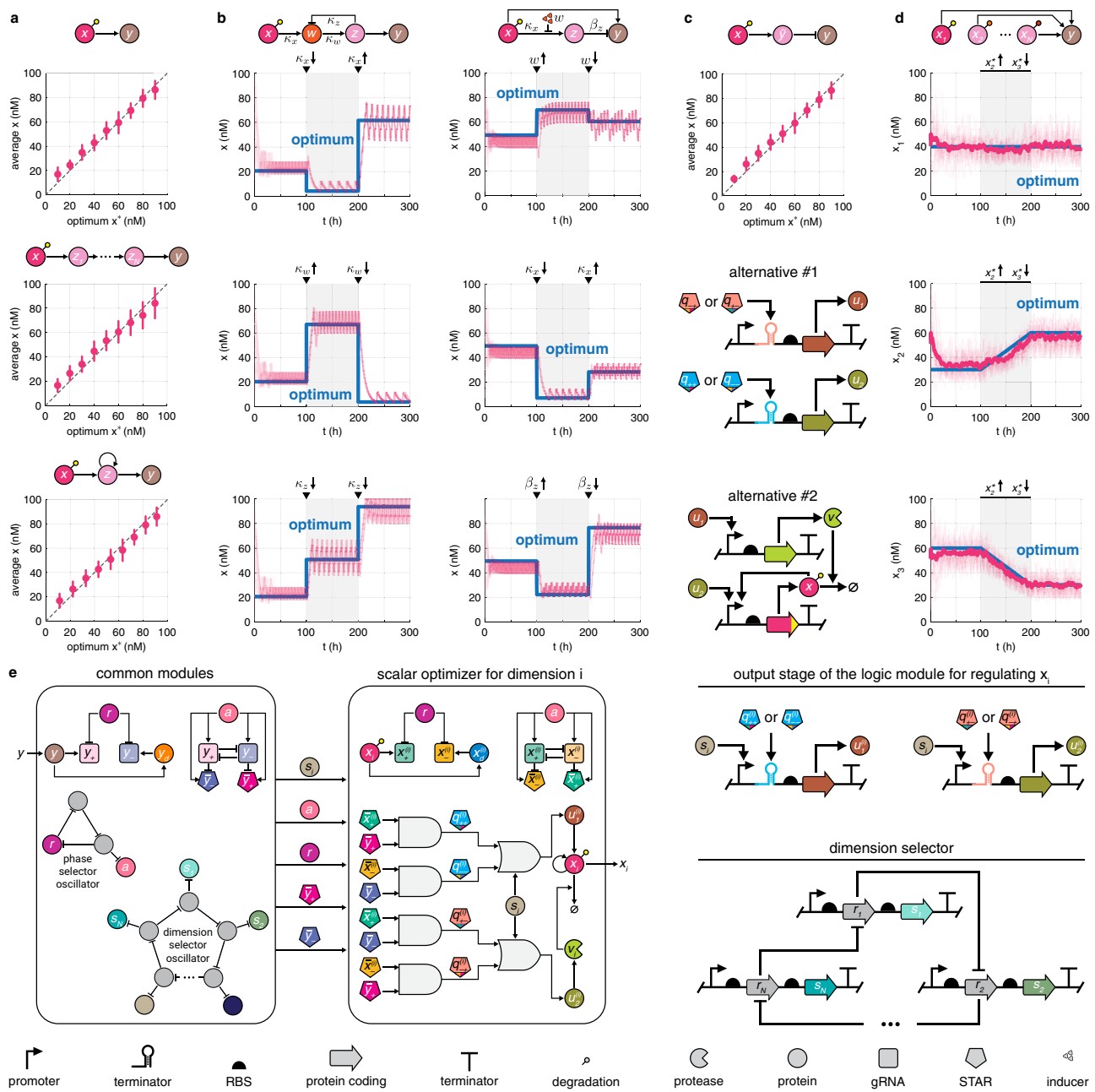

**Fig. 7 | The genetic optimizer can be successfully deployed in diverse contexts.** Detailed mathematical models and additional data are provided in Supplementary Section 3, together with simulation parameters in Supplementary Section 5. In **a**, **c**, mean and error bars denote the average of $x$ and its standard deviation, averaged over 100 independent simulations with randomly selected initial conditions. In **b**, **d**, thin red curves correspond to 30 independent closed loop trajectories with random initial points. **a** The optimizer locates the static optimum. **b** The optimizer tracks the time-varying optimum (blue) as parameters fluctuate (indicated by the arrowheads). **c** The expression of $\tilde{y}$ can be maximized by minimizing $y$. **d** The optimizer tracks the time-varying optimum (blue) even when $y$ is regulated by multiple species. The thick red curves denote the average of 30 independent simulations. **e** Genetic layout of the multi-dimensional optimizer re-using and modifying the modules originally featured in Fig. 6.

genetic optimizer can be deployed in a variety of domains to regulate cellular performance along multiple dimensions for dynamic and optimal control of microbial cell factories, to ensure the affordable and sustainable synthesis of a vast array of products ranging from biofuels to pharmaceuticals[81,82]. We anticipate the main limitation to stem from the delay between changes in the regulator $x$ and its impact on the reporter $y$, as excessive lag can lead to deteriorating performance (Supplementary Fig. 15). Additionally, while the optimizer is unable to respond to shifts that are faster than the oscillator period, it can successfully track changes that occur less rapidly (Figs. 7b, d and 8d and Supplementary Fig. 17). As a result, we expect

that the proposed implementation of the optimizer is especially well-suited in contexts where shifts in the culture environment occur slowly (on a timescale ranging from hours to days), which is typical in many fed-batch culture-based metabolic engineering applications[83–85].

The biomolecular circuit outlined in Fig. 6 is underpinned by existing synthetic biology modules (although their integration may require considerable experimental fine-tuning), and it can be successfully deployed in diverse settings as we illustrate through multiple application examples in Figs. 7 and 8 to ensure optimal closed loop performance that is robust to parameter variations and stochastic

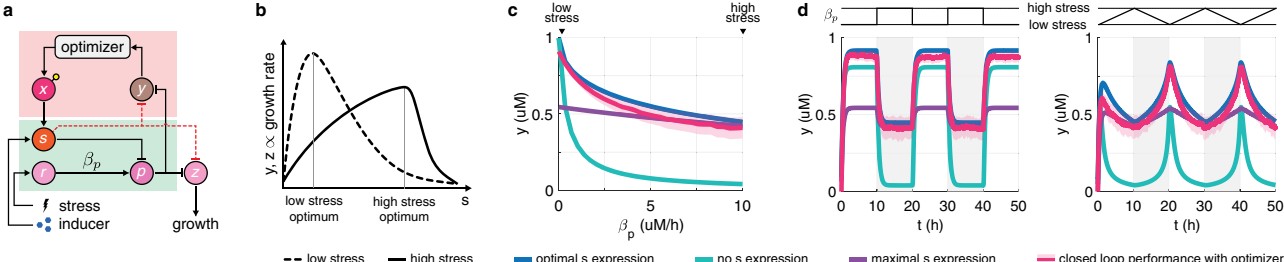

**Fig. 8 | The genetic optimizer can be deployed to maximize cellular growth rate.** Detailed mathematical model and additional data are provided in Supplementary Section 3, together with simulation parameters in Supplementary Section 5. Blue curves indicate performance when SpoTH is exogenously optimized by adjusting the inducer concentration. Green and purple curves denote trajectories with zero and maximal induction of SpoTH. Closed loop performance is evaluated in the presence of stochastic noise impacting the kinetics of all species. Cellular stress is modulated via $\beta_p$.[63] **a** Growth rate is negatively impacted by rising levels of the alarmone (p)ppGpp ($p$) as it downregulates the production of ribosomes ($z$). Cellular stress results in elevated RelA ($r$) expression, which upregulates the synthesis of (p)ppGpp via the increased production rate constant $\beta_p$. Conversely, (p)ppGpp concentration can be decreased via SpoTH ($s$) by activating its expression either exogenously[63] or by placing its promoter under the control of the regulator $x$. The dashed red flat headed arrows from SpoTH

represent the load that SpoTH expression places on ribosomes as its mRNA is translated. **b** Expression of SpoTH results in sequestration of shared cellular resources, thus the metabolic burden due to SpoTH overexpression can counteract the positive impact of (p)ppGpp removal on growth rate, resulting in a non-monotonic relationship. **c** Closed loop performance is evaluated based on 100 independent simulations with random initial conditions during the second half of each simulation by considering the average of $y$ and its standard deviation. Red curve and red shaded region denote the mean of these averages and standard deviations, respectively. **d** The optimizer successfully tracks the time-varying optimum in response to both abrupt and gradual changes in $\beta_p$ representing cellular stress. Red curves and shaded regions correspond to the mean and standard deviation of trajectories considering 100 independent simulations with random initial conditions. For individual trajectories, see Supplementary Fig. 17.

noise, without requiring any application-specific tuning of its parameters. Considering the parameter values summarized in Supplementary Table 1 and their typical ranges discussed in Supplementary Section 2.2, the concrete molecular implementation of the whole integrated system in Fig. 6 is expected to have a bioenergetic cost similar to genetic circuits[86] that have already been implemented without negatively impacting growth rate (Supplementary Section 4). Finally, as the coarse-grained mechanistic model underpinning our approach reveals fundamental principles that ensure correct functioning, we envision that other realizations of the proposed genetic optimizer will be implemented by incorporating synthetic components from a diverse set of kits similar to recent approaches[52,87], combining plasmid copy number control[88], transcriptional regulation[42,43], post-translational modification[89], RNA-based interactions[36,71], and especially fast phosphotransfer processes[90].

Biomolecular controllers offer a promising avenue towards robust and optimal gene regulation, an essential feature of complex synthetic systems. The rationally designed pathway-independent optimizer presented here builds on ideas rooted in control theory to implement a genetic feedback module that ensures convergence to, and tracking of, the time-varying optimum even in the presence of disturbances and stochastic noise. The blueprint of this module can be realized in a versatile fashion, relying on the large variety of already existing bioengineering parts.

## Methods
### Deterministic simulations
Data corresponding to deterministic dynamics were obtained using the ode45 solver of MATLAB.

### Stochastic simulations
Data corresponding to stochastic dynamics were obtained as follows. Define the discretization $0 = t_0 < t_1 < \cdots < t_N = T$ over the time interval $[0, T]$ with $t_i = i\delta$ where $\delta = T/N$ is the stepsize, and introduce $\Delta W_n = W(t_{n+1}) - W(t_n)$ where $W$ is a $d$-dimensional standard Wiener process. The solution of the $d$-dimensional system of (Itô) stochastic differential equations

$$dX(t) = \mu(X(t), \Theta)dt + \sigma(X(t), \Theta)dW(t), \qquad X(0) = x_0,$$

with $\mu \in \mathbb{R}^d$ drift and $\sigma \in \mathbb{R}^{d \times d}$ diffusion coefficient according to the multi-dimensional Euler–Maruyama scheme[91,92] is given by

$$X_{n+1} = X_n + \delta\mu(X_n, \Theta) + \sigma(X_n, \Theta)\Delta W_n, \qquad X_0 = x_0,$$

where $X_j$ denotes the numerical approximation of $X(t_j)$.

### Data collection
Simulation data were generated using MATLAB (version R2022b).

### Data analysis
Simulation data were analyzed and plotted using MATLAB (version R2022b). Figures were created using Adobe Illustrator (version 26.5). Overleaf (version v2) online LaTeX editor was used to prepare and compile the manuscript.

### Reporting summary
Further information on research design is available in the Nature Portfolio Reporting Summary linked to this article.

## Data availability
Source data are provided with this paper.

## Code availability
The manuscript does not rely on custom mathematical algorithms or software. Simulation data were generated and analyzed as described in the "Methods" by considering the models detailed in the Supplementary Information using built-in MATLAB functions. The MATLAB code for implementing and deploying the optimizer for the examples featured in the manuscript, along with instructions on how to modify it for other application examples, are publicly available at https://github.com/qbionet/genetic-optimizer. Additional information is available from the corresponding author upon request.

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

## Acknowledgements

This work was supported by research funds from New York University Abu Dhabi and by DARPA HR0011-16-2-0044. The authors would like to thank Adam P. Arkin for posing the genetic optimization problem and facilitating its initial study under the DARPA award, and Shivang H. N. Joshi for helpful comments about the parameter values used in this study.

## Author contributions

A.G., A.M., and M.A. contributed to the conception and design of the genetic optimizer. A.G. developed the mathematical model of the genetic optimizer with inputs from A.M. and M.A. A.G. analyzed the mathematical model, designed the molecular implementation, collected and analyzed the simulation data, created the figures, and drafted the paper and the revision. A.G., A.M., and M.A. jointly edited the manuscript.

## Competing interests

The authors declare no competing interests.
