## [Peer Review File · Nature Communications]

Reviewers' Comments:

Reviewer #1:

Remarks to the Author:

The manuscript by Gyorgy et al. describes a method to construct synthetic genetic feedback optimizer by applying control theory and utilizing a number of different synthetic genetic elements. They propose a genetic feedback optimizer by utilizing gradient descent approach via application of appropriate transformations required for chemical species (e.g., non-negativeness). They break down the required steps as delay module, comparator module, and logic module, and discuss the extension to multi-dimensional cases and potential implementation at the molecular level. The developed theory and simulation results are well explained. However, the details on molecular implementation is lacking.

Specifically, the authors need to address the following.

1. The key limitation of this manuscript is lack of details on molecular implementation. The proposed scheme requires integration of multiple synthetic genetic elements -- oscillator, bistable switch, and logic elements -- each even individually can be quite challenging to properly construct and characterize. The authors need to provide experimentally realizable scheme for the whole integrated system rather than a hypothetical scheme for individual pieces as proposed in the supplement. Especially problematic is the oscillator period. Experimentally demonstrated synthetic oscillators typically have periods close to 1 hour and this by no means would be sufficiently fast for the proposed control scheme. The authors need to provide an alternative scheme or propose a experimentally feasible scheme to boost speed of oscillators. In the supplement, the authors cite a theoretical paper to get parameters. They need to use experimental papers to get biologically plausible parameters.

Other minor points include the following:

1. in page 4, we then we briefly -> we then briefly
2. in page 10, Figure 6d -> Figure 3d
3. Reference 42 and reference 69 are identical.

Reviewer #2:

Remarks to the Author:

Gyorgy et al. proposed a design strategy of genetic feedback networks that act as an online optimizer of a certain user-defined performance function of the cell such as the cell growth. The authors proposed a cascade system of three modules that adaptively generates the input (control) signaling molecules (u) based only on the concentrations of intracellular molecules (x) and the values of the performance function ($y=F(x)$). Roughly speaking, these modules mimic the mechanism of the gradient decent/ascent method in computational optimization, where the target variable x is "updated" along the direction of the steepest slope of $y=F(x)$ using the input u . In the paper, the authors introduced mathematical models of the modules and performed computational simulations of the models to verify this idea.

The proposed concept is relevant to the engineering of synthetic genetic network, but the reviewer thinks that the feasibility of the proposed concept is questionable (please see the item-by-item comments below). This is partly because the paper focuses on the abstract mathematical concept, and its realization is discussed only qualitatively (without experimentally verified parameters etc.) in the short section at the end of the paper and in the supplementary text. Overall, the paper needs more quantitative discussions on the feasibility of the concept based on experimental data in literatures. These include the feasibility of tuning parameters (for example, α and c), realistic time-scales of the reactions, and the effect of stochastic noise etc.

1. What is the specific physical entity of y that can be measured by the optimizer network in eq. (3). In line 107, the paper raised the growth rate as an example, but the reviewer wonders how the growth rate can be "sensed" by the cellular reactions and processed to the three modules. It would be useful to discuss what specifically can (or cannot) be optimized by the proposed optimizer.

2. Line 110-113: the paper said that the proposed idea can be generalized to a broad class of

systems including the time-varying parameter case, but the reviewer wonders (i) how broad the class is, and (ii) whether the time-varying case can actually be optimized by the proposed optimizer. It seems that the time-varying case is not discussed in the paper. In particular, perfect tracking of the (moving) optimum values seem difficult since the optimizer has a lag to generate the input signal u . Please provide concrete reasonings or conditions that the optimizer can track the time-varying optimal values, and show "the class" that is considered in this paper.

3. Line 168: the paper assumes that c is a binary variable for simplicity, but the reviewer suspects that this is too much simplification since the time-scale of the genetic oscillator is almost the same as the comparator dynamics. Would it still function as a comparator if the oscillator dynamics (with some realistic parameters from literatures) are included in the model?

4. In addition to the above comments, experimentally, the toggle switch and the oscillator modules in ref. 48 and 49 exhibited stochastic gene expression kinetics. These stochastic effects would largely alter the single-cell reaction kinetics from what is simulated in the paper. Please provide the reasoning that these stochastic effects can be ignored for the proposed modules and/or discuss the effect of the stochastic noise on the modules.

5. In Fig. 5 and 6, it would be helpful to show a more specific example of genetic networks with realistic parameters. The reviewer particularly wonders whether the closed loop control can be achieved within a realistic time-scale of experiments since the optimizers consist of many modules that require gene expression, each of which generates a lag. Moreover, the reviewer is concerned that the optimizer itself would consume many resource molecules.

Response to Referees

We thank the Editor and the Reviewers for their helpful and constructive feedback. Addressing these has enabled us to improve our work considerably by demonstrating that the optimizer can be successfully deployed in a variety of contexts when relying on an implementation combining existing synthetic biology parts and components together with mass action kinetics-based dynamics and parameter values typical in *E. coli*. Accordingly, the manuscript has been revised and its scope has been substantially expanded in four major ways:

1. **In Fig. 6, we outline a concrete molecular implementation of the whole integrated system to demonstrate that our optimizer can be realized by combining existing synthetic biology modules.** This realization relies on already available genetic parts and components, in particular: (i) protein-based transcriptional control; (ii) inducible degradation via the *M. florum* Lon protease and *ssrA* tag; (iii) a repressilator-based oscillator; (iv) CRISPRi-based toggle switches; and (v) STAR-based logic gates. The corresponding mass action kinetics-based dynamical model is included in Supplementary Section 2.1.
2. **In Fig. 7–Fig. 8, we illustrate that our optimizer successfully locates and tracks the (static or time-varying) optimum in diverse contexts when relying on typical parameter values considering the implementation in Fig. 6.** The application examples include maximization and minimization when the regulator x directly affects the production of the reporter y , alongside with scenarios when its impact is realized through more complex pathways involving activation and repression, autoregulation, regulatory cascades, feedforward and feedback loops, and even multidimensional optimization. Across all examples, parameters of the optimizer are kept fixed, as summarized in Supplementary Table 1. These values lie within the typical range of biophysical parameters observed in *E. coli*, as we discuss in Supplementary Section 2.2 considering over 80 experimental papers. For instance, parameters of the repressilator-based oscillator module were selected to obtain a period length of approximately 2 hours (Fig. 7), well within the range of already existing experimental realizations (1–4).
3. **In Supplementary Fig. 8–Fig. 20, we verify that closed loop performance is robust to stochastic noise and perturbations in the parameter values of both our optimizer and the regulated pathway.** Perturbations range from changes in the parameters affecting the cycle length of the oscillator to the shape of the objective function. The optimizer successfully locates and tracks the (static or time-varying) optimum even in the presence of significant stochastic noise impacting single-cell reaction kinetics that causes completely absent periods of the oscillator and random switches between the two stable equilibria of the toggle switches.

4. **In the Discussion, we highlight that our optimizer can be readily integrated with existing pathways and genetically encoded biosensors to ensure its successful deployment in a variety of settings, which we illustrate in Fig. 8 through growth rate regulation.** In Fig. 8 we present a potential pathway to regulate growth rate in the presence of cellular stress by leveraging the strong inverse relationship between the alarmone (p)ppGpp and both RNA synthesis and growth. This is achieved by interfacing a recently developed exogenous growth rate controller with the optimizer module to modulate the expression of SpoTH for controlled (p)ppGpp hydrolysis. Sensing of growth rate is achieved by placing the reporter protein under the control of an *rrn* P1 promoter to create a proxy signal for ribosome production and growth rate, as rRNA synthesis is the rate-limiting step in ribosome production and thus of growth. This example highlights how existing regulatory solutions based on exogenous control can be enhanced by combining them with the optimizer without compromising performance or requiring any modification to already developed pathways. As we further detail in the Discussion, synthesis of the reporter can be made responsive to a multitude of signals and analytes relying on the vast collection of protein-based and RNA-based biosensors to regulate cellular performance along multiple dimensions for dynamic and optimal control of microbial cell factories in diverse settings.

Additional minor revisions include (i) streamlining the presentation of the results originally featured in the manuscript; (ii) modification of the figures to conform to journal guidelines (e.g., font size); and (iii) removal of Fig. 5a and its inclusion in the Supplementary Information as Supplementary Fig. 6a. In what follows, we turn to Reviewer comments (highlighted in blue) and detail how we addressed them one-by-one. In addition to the revised manuscript, we have also uploaded a copy indicating all the changes (generated using the `latexdiff` package: added text is blue, discarded text is red).

Reviewer #1

The manuscript by Gyorgy *et al.* describes a method to construct synthetic genetic feedback optimizer by applying control theory and utilizing a number of different synthetic genetic elements. They propose a genetic feedback optimizer by utilizing gradient descent approach via application of appropriate transformations required for chemical species (e.g., non-negativeness). They break down the required steps as delay module, comparator module, and logic module, and discuss the extension to multi-dimensional cases and potential implementation at the molecular level. The developed theory and simulation results are well explained. However, the details on molecular implementation is lacking.

We thank the Reviewer for their time spent reviewing our manuscript, and for their constructive feedback and suggestions that helped us not only improve the results presented in the original submission, but also sparked novel lines of research inquiry substantially expanding the scope of our work. These include the concrete molecular implementation of the whole integrated system (Fig. 6) with the corresponding mass action kinetics relying on parameter values typical in *E. coli* (Supplementary Section 2), alongside with a diverse set of application examples demonstrating that the optimizer ensures successful tracking of the unknown optimum (Fig. 7–Fig. 8), without requiring any application-specific tuning of its parameters. In what follows, we address the helpful comments that we have received one-by-one.

Specifically, the authors need to address the following. The key limitation of this manuscript is lack of details on molecular implementation. The proposed scheme requires integration of multiple synthetic genetic elements – oscillator, bistable switch, and logic elements – each even individually can be quite challenging to properly construct and characterize. The authors need to provide experimentally realizable scheme for the whole integrated system rather than a hypothetical scheme for individual pieces as proposed in the supplement. Especially problematic is the oscillator period. Experimentally demonstrated synthetic oscillators typically have periods close to 1 hour and this by no means would be sufficiently fast for the proposed control scheme. The authors need to provide an alternative scheme or propose a experimentally feasible scheme to boost speed of oscillators. In the supplement, the authors cite a theoretical paper to get parameters. They need to use experimental papers to get biologically plausible parameters.

To address this shortcoming, we now include a concrete molecular implementation of the whole integrated system in Fig. 6 that is experimentally realizable by combining genetic parts and modules that are already available (though their integration may require considerable experimental fine-tuning, which we now acknowledge in the Discussion). In particular: (i) protein-based transcriptional control; (ii) inducible degradation via the *M. florum* Lon protease and *ssrA* tag operating independently from and orthogonally to the host degradation pathways; (iii) a repressilator-based oscillator given its widespread use in a variety of experimental settings; (iv) CRISPRi-based toggle switches; and (v) STAR-based logic gates due to their highly programmable and orthogonal nature. Complementing the description of each module included in the section titled “The optimizer can be implemented with existing synthetic biology modules,” we provide additional technical details in Supplementary Section 2. In Supplementary Section 2.1, we present the mass action kinetics-based mathematical model underpinning each of the modules, together with a detailed discussion of the typical range of model parameters in Supplementary Section 2.2 considering over 80 experimental papers. Based on this, we summarize the parameters of the optimizer and their selected values in Supplementary Table 1 that are kept fixed across all application examples featured in Fig. 7–Fig. 8 (and throughout the Supplementary Information).

These include maximization and minimization when the regulator x directly affects the production of the reporter y , alongside with scenarios when its impact is realized through more complex pathways involving activation and repression, autoregulation, regulatory cascades, feedforward and feedback loops, and even multidimensional optimization. Across all examples, the genetic optimizer (with the concrete molecular implementation of the whole integrated system in Fig. 6, parameterized as in Supplementary Table 1) ensures successful tracking of the unknown (static or time-varying) optimum. We also demonstrate in Supplementary Fig. 8–Fig. 20 that closed loop performance is robust to considerable perturbations, ranging from changes in the parameters that affect the cycle length of the oscillator to the shape of the objective function. Similarly, the optimizer locates and tracks the (time-varying) optimum in the presence of significant stochastic noise impacting single-cell reaction kinetics, even when noise intensity is sufficient to cause completely absent periods of the oscillator and random switches between the two stable equilibria of the toggle switches. These results highlight that the proposed genetic optimizer module can be successfully deployed in a variety of settings by relying on the vast collection of protein-based and RNA-based biosensors to regulate cellular performance along multiple dimensions for dynamic and optimal control of microbial cell factories. This is now detailed in the Discussion together with an illustration focusing on growth rate control in Fig. 8 and an assessment of the bioenergetic cost of the optimizer considering two reference circuits that have already been implemented and tested.

Finally, we replaced the previously featured activator-repressor clock with the repressilator to serve as the oscillator in the comparator module given its widespread use in a variety of experimental settings. We selected its parameters to obtain a period length of approximately 2 hours (Supplementary Fig. 7) to ensure that it falls well within the range of experimentally demonstrated synthetic oscillators with periods spanning the range from 13 min (2) to approximately 10 h (3). As we illustrate through multiple application examples in Fig. 7–Fig. 8, such an oscillator is sufficiently fast for the proposed control scheme considering the typical range of the degradation rate constants and growth rate in *E. coli* that establish the timescale of both the slower protein-based and faster RNA-based components of the optimizer, as well as the timescale of the pathway that needs to be regulated. In particular, we demonstrate that the optimizer (with the concrete molecular implementation of the whole integrated system in Fig. 6, parameterized as in Supplementary Table 1) can successfully locate the optimum (Fig. 7ac, Fig. 8c), re-locate it subsequent to abrupt changes in a variety of parameters (Fig. 7b, Fig. 8d), and even track it as its time-varying location continuously shifts (Fig. 7d, Fig. 8d) at varying rates (Supplementary Fig. 17), without requiring any application-specific tuning of its parameters. Regarding the timescales, we expect the main limitation to stem from the delay between changes in the regulator x and its impact on the reporter y : data in Supplementary Fig. 15 demonstrate that excessive lag can lead to deteriorating performance, which is now explicitly stated in the Discussion. Importantly, however, considerable delay can be tolerated without compromising closed loop behavior, as we illustrate for a regulatory cascade comprising four activator transcription factors each with approximately 15 min half-life (Supplementary Fig. 15). Additionally, while the optimizer can track changes in the optimum that are slower than the oscillator period (Fig. 7bd, Fig. 8d, Supplementary Fig. 17), it is not able respond to shifts that occur more rapidly (this is now also acknowledged in the Discussion).

Other minor points include the following:

1. in page 4, we then we briefly → we then briefly

This typo has been corrected.

2. in page 10, Figure 6d → Figure 3d

This typo has been corrected.

3. Reference 42 and reference 69 are identical.

The duplicate reference has been removed.

Reviewer #2

Gyorgy *et al.* proposed a design strategy of genetic feedback networks that act as an online optimizer of a certain user-defined performance function of the cell such as the cell growth. The authors proposed a cascade system of three modules that adaptively generates the input (control) signaling molecules (u) based only on the concentrations of intracellular molecules (x) and the values of the performance function ($y = F(x)$). Roughly speaking, these modules mimic the mechanism of the gradient decent/ascent

method in computational optimization, where the target variable x is “updated” along the direction of the steepest slope of $y = F(x)$ using the input u . In the paper, the authors introduced mathematical models of the modules and performed computational simulations of the models to verify this idea.

The proposed concept is relevant to the engineering of synthetic genetic network, but the reviewer thinks that the feasibility of the proposed concept is questionable (please see the item-by-item comments below). This is partly because the paper focuses on the abstract mathematical concept, and its realization is discussed only qualitatively (without experimentally verified parameters etc.) in the short section at the end of the paper and in the supplementary text. Overall, the paper needs more quantitative discussions on the feasibility of the concept based on experimental data in literatures. These include the feasibility of tuning parameters (for example, α and c), realistic time-scales of the reactions, and the effect of stochastic noise etc.

We thank the Reviewer for their time spent reviewing our manuscript, and for their constructive feedback and suggestions that helped us not only improve the results presented in the original submission, but also sparked novel lines of research inquiry substantially expanding the scope of our work. These include the concrete molecular implementation of the whole integrated system relying on already available parts and components (Fig. 6), alongside with the parameterization of the corresponding mass action kinetics relying on biologically realistic values (Supplementary Section 2), together with a diverse set of application examples demonstrating that the optimizer ensures successful tracking of the unknown optimum (Fig. 7–Fig. 8), without requiring any application-specific tuning of its parameters. We also demonstrate in Supplementary Fig. 8–Fig. 20 that closed loop performance is robust to considerable perturbations, ranging from changes in the parameters that affect the cycle length of the oscillator to the shape of the objective function. Similarly, the optimizer locates and tracks the (time-varying) optimum in the presence of significant stochastic noise impacting single-cell reaction kinetics (Fig. 8), even when noise intensity is sufficient to result in completely absent periods of the oscillator, and in random switches between the two stable equilibria of the toggle switches (Supplementary Fig. 18). Finally, in the Discussion we highlight that the genetic optimizer can be successfully deployed in a variety of settings by relying on the vast collection of protein-based and RNA-based biosensors to regulate cellular performance along multiple dimensions for dynamic and optimal control of microbial cell factories, which we illustrate through growth rate control (Fig. 8). In what follows, we address the helpful comments that we have received one-by-one.

1. What is the specific physical entity of y that can be measured by the optimizer network in eq. (3). In line 107, the paper raised the growth rate as an example, but the reviewer wonders how the growth rate can be “sensed” by the cellular reactions and processed to the three modules. It would be useful to discuss what specifically can (or cannot) be optimized by the proposed optimizer.

To address this shortcoming, we expanded the Discussion in two ways.

First, we present in Fig. 8 a potential pathway to regulate growth rate via the alarmone (p)ppGpp by interfacing it with the concrete molecular implementation of the whole integrated system in Fig. 6. In particular, the growth rate controller developed in (5) and outlined in Fig. 8a can be deployed to exogenously adjust growth rate via SpoTH. While expression of SpoTH has a positive impact on growth rate at moderate levels as a result of (p)ppGpp hydrolysis, its overexpression can result in

considerable metabolic burden (e.g., due to ribosome sequestration), thus giving rise to the non-monotonic relationship between SpoTH expression and growth rate observed in (5) and illustrated in Fig. 8b. By placing the reporter protein y under the control of an *rrn* P1 promoter, we create a proxy signal for ribosome production and growth rate, and ensure that the optimizer indirectly maximizes growth rate via y , leveraging that rRNA synthesis is the rate-limiting step in ribosome production, and thus of growth. Data presented in Fig. 8cd demonstrate that (i) optimal regulation of SpoTH, and thus of (p)ppGpp expression can significantly mitigate the negative impact of cellular stress on growth rate; (ii) performance with the the optimizer approaches the theoretical optimum that can be achieved by careful exogenous tuning of SpoTH expression; and (iii) closed loop behavior is robust to considerable stochastic noise and parameter variations (Supplementary Fig. 16–Fig. 20).

Second, as illustrated by the above example, we discuss how already existing regulatory solutions based on exogenous control can be enhanced by combining them with the proposed optimizer without compromising performance or requiring any modification to already developed pathways. Both the regulator x and the reporter y can be implemented using widely available transcription factors, and the synthesis of y can be made responsive to a multitude of signals and analytes relying on the vast collection of protein-based and RNA-based biosensors (e.g., references (71)–(74) feature dozens of common genetically encoded biosensors that should aid interested readers in further exploration). For instance, transcription of y can be modulated by expressing it from the cognate promoter of a ligand inducible transcription factor, whereas metabolite-sensing riboswitches can be deployed to directly impact translation initiation of y . Thus, the optimizer can be easily interfaced with the continuously expanded and already vast collection of biosensors responsive to common metabolites ranging from precursors to biosynthetic pathways through products from secondary metabolism to various environmental signals. Consequently, we expect the genetic optimizer to be deployed in diverse settings to regulate cellular performance along multiple dimensions for dynamic and optimal control of microbial cell factories to ensure the affordable and sustainable synthesis of a vast array of products.

2. Line 110-113: the paper said that the proposed idea can be generalized to a broad class of systems including the time-varying parameter case, but the reviewer wonders (i) how broad the class is, and (ii) whether the time-varying case can actually be optimized by the proposed optimizer. It seems that the time-varying case is not discussed in the paper. In particular, perfect tracking of the (moving) optimum values seem difficult since the optimizer has a lag to generate the input signal u . Please provide concrete reasonings or conditions that the optimizer can track the time-varying optimal values, and show “the class” that is considered in this paper.

To address this issue, we now feature several application examples in Fig. 7–Fig. 8 to demonstrate that the optimizer can successfully locate and track the optimum in diverse contexts, both when the location of the optimum changes abruptly and when it shifts gradually.

These include maximization and minimization when the regulator x directly affects the production of y , alongside with scenarios when its impact is realized through more complex pathways involving activation and repression, autoregulation, regulatory cascades, feedforward and feedback loops, and even multidimensional optimization. We demonstrate that the optimizer (with the concrete molecular implementation of the whole integrated system in Fig. 6, parameterized

as in Supplementary Table 1) can successfully locate the optimum (Fig. 7ac, Fig. 8c), re-locate it subsequent to abrupt changes in a variety of parameters (Fig. 7b, Fig. 8d), and even track it as its time-varying location continuously shifts (Fig. 7d, Fig. 8d) at varying rates (Supplementary Fig. 17), without requiring any application-specific tuning of its parameters.

We also demonstrate in Supplementary Fig. 8–Fig. 20 that closed loop performance is robust to considerable perturbations, ranging from changes in the parameters that affect the cycle length of the oscillator to the shape of the objective function. Similarly, the optimizer locates and tracks the (time-varying) optimum in the presence of significant stochastic noise impacting single-cell reaction kinetics (Fig. 8), even when noise intensity is sufficient to result in completely absent periods of the oscillator, and in random switches between the two stable equilibria of the toggle switches.

Importantly, this performance is ensured when considering the concrete molecular implementation of the whole integrated system in Fig. 6 (we present the mathematical model underpinning each of the modules in Supplementary Section 2.1), and when keeping the parameter values of the optimizer fixed across all examples as indicated in Supplementary Table 1 (detailed discussion of the typical range of model parameters are provided in Supplementary Section 2.2 based on over 80 experimental papers).

These results demonstrate that the proposed optimizer can be successfully deployed for a variety of pathways that are common in synthetic biology applications (we have removed the phrase “broad class of systems” as we are not providing a precise mathematical definition of such class). We expect the main limitation to stem from the delay between changes in the regulator x and its impact on the reporter y : data in Supplementary Fig. 15 demonstrate that excessive lag can lead to deteriorating performance, which is now explicitly stated in the Discussion. Importantly, however, considerable delay can be tolerated without compromising closed loop behavior, as we illustrate for a regulatory cascade comprising four activator transcription factors each with approximately 15 min half-life (Supplementary Fig. 15). Additionally, while the optimizer can track changes in the optimum that are slower than the oscillator period (Fig. 7bd, Fig. 8d, Supplementary Fig. 17), it is not able respond to shifts that occur more rapidly (this is now also acknowledged in the Discussion).

3. Line 168: the paper assumes that c is a binary variable for simplicity, but the reviewer suspects that this is too much simplification since the time-scale of the genetic oscillator is almost the same as the comparator dynamics. Would it still function as a comparator if the oscillator dynamics (with some realistic parameters from literatures) are included in the model?

To demonstrate that this simplification is not central to our findings, in Fig. 6 we first include a concrete molecular implementation of the integrated system (including the oscillator) relying on existing synthetic biology parts and modules. In this realization of the optimizer, we selected the repressilator to serve as the oscillator in the comparator module due to its widespread use in a variety of experimental settings. Complementing the description of each module included in the section titled “The optimizer can be implemented with existing synthetic biology modules,” we provide additional technical details in Supplementary Section 2. In Supplementary Section 2.1 we present the mass action kinetics-based mathematical model underpinning the dynamics of each module, together with a detailed discussion of the typical range of model parameters in Supple-

mentary Section 2.2 considering over 80 experimental papers. Based on this, we summarize the parameters of the optimizer and their selected values in Supplementary Table 1. Regarding the repressilator, we chose its parameters to obtain a period length of approximately 2 hours (Supplementary Fig. 7), well within the range of already existing experimental realizations (1, 3, 4).

As we detail in our response to point #2 above raised by the Reviewer, we now feature several application examples in Fig. 7–Fig. 8 to demonstrate that the optimizer can locate and track the optimum in diverse contexts, both when the location of the optimum changes abruptly and when it shifts gradually. In all these examples, we rely on the mass action kinetics-based dynamics of the whole integrated system (Supplementary Section 2.1) featured in Fig. 6 (including the repressilator-based oscillator, instead of the binary variable c), and parameters of the optimizer are kept fixed, as detailed in Supplementary Table 1. These results verify that the optimizer can be successfully interfaced with a variety of pathways that are common in synthetic biology applications, without requiring any application-specific tuning of its parameters. We also illustrate in Supplementary Fig. 8–Fig. 20 that closed loop performance is robust to considerable perturbations and stochastic noise. Taken together, these results demonstrate that the optimizer can be successfully deployed in a variety of contexts when relying on (i) an implementation combining existing synthetic biology parts and components; (ii) mass action kinetics-based dynamics; and (iii) biologically plausible parameter values typical in *E. coli*.

4. In addition to the above comments, experimentally, the toggle switch and the oscillator modules in ref. 48 and 49 exhibited stochastic gene expression kinetics. These stochastic effects would largely alter the single-cell reaction kinetics from what is simulated in the paper. Please provide the reasoning that these stochastic effects can be ignored for the proposed modules and/or discuss the effect of the stochastic noise on the modules.

In a new example focusing on growth rate control (Fig. 8), we replaced the ordinary differential equations with stochastic differential equations governing the dynamics of the concrete molecular implementation of the whole integrated system outlined in Fig. 6. This way we included stochastic noise for each module, not only for the oscillator and the toggle switches (Supplementary Section 3). As we demonstrate in Fig. 8, the optimizer locates and tracks the unknown and time-varying optimum even when noise intensity is sufficient to result in completely absent periods of the oscillator, and in random switches between the two stable equilibria of the toggle switches (Supplementary Fig. 16). These results verify that the optimizer can be successfully deployed not only in the absence of noise, but also when stochastic effects have considerable impact on single-cell reaction kinetics, and thus on the behavior of the functional modules (Fig. 8, Supplementary Fig. 16–Fig. 20).

5. In Fig. 5 and 6, it would be helpful to show a more specific example of genetic networks with realistic parameters. The reviewer particularly wonders whether the closed loop control can be achieved within a realistic time-scale of experiments since the optimizers consist of many modules that require gene expression, each of which generates a lag. Moreover, the reviewer is concerned that the optimizer itself would consume many resource molecules.

To verify that closed loop control can be achieved with a realistic realization, in Fig. 6 we include a concrete molecular implementation of the whole integrated system relying on existing synthetic

biology parts and modules (though their integration may require considerable experimental fine-tuning, which we now acknowledge in the Discussion). In particular: (i) protein-based transcriptional control; (ii) inducible degradation via the *M. florum* Lon protease and *ssrA* tag operating independently from and orthogonally to the host degradation pathways; (iii) a repressilator-based oscillator given its widespread use in a variety of experimental settings; (iv) CRISPRi-based toggle switches; and (v) STAR-based logic gates due to their highly programmable and orthogonal nature. As a result, parameters of the implementation can be tuned by relying on standard synthetic biology tools in a straightforward fashion by adjusting transcriptional, translational, degradation, and dissociation rate constants (Supplementary Section 2.2), e.g., to modulate the period of the oscillator (Supplementary Fig. 19–Fig. 20).

Complementing the description of each module included in the section titled “The optimizer can be implemented with existing synthetic biology modules,” we provide additional technical details in Supplementary Section 2. In Supplementary Section 2.1 we present the mass action kinetics-based mathematical model underpinning the dynamics of each module, together with a detailed discussion of the typical range of model parameters in Supplementary Section 2.2 considering over 80 experimental papers. Based on this, we summarize the parameters of the optimizer and their selected values in Supplementary Table 1 that are kept fixed across all application examples featured in Fig. 7–Fig. 8. As we detail in our response to point #2 above raised by the Reviewer, data in Fig. 7–Fig. 8 highlight that the optimizer can locate and track the optimum in diverse contexts without requiring any application-specific tuning of its parameters, both when the location of the optimum changes abruptly and when it shifts gradually. We also illustrate in Supplementary Fig. 8–Fig. 20 that closed loop performance is robust to considerable perturbations and stochastic noise. Taken together, these results demonstrate that the optimizer can be successfully deployed in a variety of contexts when relying on (i) an implementation combining already existing synthetic biology parts and components; (ii) mass action kinetics-based dynamics; and (iii) biologically plausible parameter values typical in *E. coli*.

Finally, to assess the bioenergetic cost of the optimizer, we consider two reference circuits that have already been implemented and tested. As we outline in Supplementary Section 4, we estimate that the metabolic burden of the concrete molecular implementation of the whole integrated system outlined in Fig. 6 when considering typical parameter values (Supplementary Section 2.2) is comparable to these reference circuits. While the optimizer comprises more components than these modules, over half of these parts are RNA-based with reduced bioenergetic cost when compared to their protein-based counterparts (as growth is primarily limited by the availability of translational resources). Since the reference circuits had no negative impact on cellular growth rate, we thus expect that the proposed optimizer can be deployed without significantly affecting growth rate (now included in the Discussion).

References

1. Elowitz, M. B. & Leibler, S. A synthetic oscillatory network of transcriptional regulators. *Nature* **403**, 335–338 (2000).
2. Stricker, J. *et al.* A fast, robust and tunable synthetic gene oscillator. *Nature* **456**, 516–519 (2008).
3. Potvin-Trottier, L., Lord, N. D., Vinnicombe, G. & Paulsson, J. Synchronous long-term oscillations in a synthetic gene circuit. *Nature* **538**, 514–517 (2016).
4. Riglar, D. T. *et al.* Bacterial variability in the mammalian gut captured by a single-cell synthetic oscillator. *Nature Communications* **10**, 4665 (2019).
5. Barajas, C., Huang, H.-H., Gibson, J., Sandoval, L. & Vecchio, D. D. Feedforward growth rate control mitigates gene activation burden. *Nature Communications* **13**, 7054 (2022).

Reviewers' Comments:

Reviewer #1:

Remarks to the Author:

The authors addressed the review comments by including example circuits with biologically plausible parameters and improved the manuscript. There are a couple of points that the authors could address further.

1. For the comparator module, only the repressor regulates the expression state of $x+$, $x-$, $y+$, $y-$ in phase 1. It would be good to indicate that the repressor level should be quite low for reliable operation of comparator in one of the the phase or propose some way to address leaky expression commonly found in biological circuits.
2. The timescale of controller is more realistic with the new parameter sets and it indicates that the process should have very slow changes in the culture environments (\sim hrs to days) for optimal use of the controller. It would be good to compare with typical experimental timescale with practical applications such as fed-batch culture in metabolic engineering to see if this approach can be applied.
3. It is not clear whether the extended discussion on Figure 8 should be in the discussion section.

Reviewer #2:

Remarks to the Author:

All of my major concerns were addressed in the revised manuscript.

In particular, the study was extended to show biological plausibility of the proposed scheme. The revised version shows examples of biological implementation of the proposed molecular optimizer. Simulations for these optimizers were performed using realistic parameters in literature, and robustness to stochastic noise was also discussed. The reviewer thinks that these extensive computational simulations are sufficient to support the main claim of the paper.

Response to Referees

We thank the Editor and the Reviewers for their constructive feedback that enabled us to improve our work and significantly expand the scope of the manuscript throughout the review process. Here, we detail how we addressed the final comments that we have received.

Reviewer #1

The authors addressed the review comments by including example circuits with biologically plausible parameters and improved the manuscript. There are a couple of points that the authors could address further.

We thank the Reviewer for their constructive feedback and suggestions that helped us not only improve the results presented in the original submission, but also sparked novel lines of research inquiry significantly expanding the scope of our work throughout review process.

1. For the comparator module, only the repressor regulates the expression state of x_+ , x_- , y_+ , y_- in phase 1. It would be good to indicate that the repressor level should be quite low for reliable operation of comparator in one of the the phase or propose some way to address leaky expression commonly found in biological circuits.

We now explicitly state this requirement when discussing the implementation of the comparator module in the section titled “Implementation with existing synthetic biology modules,” together with an available experimental technique that ensures stringent multi-level control of gene expression to reduce leaky expression commonly found in biological circuits.

2. The timescale of controller is more realistic with the new parameter sets and it indicates that the process should have very slow changes in the culture environments (\sim hrs to days) for optimal use of the controller. It would be good to compare with typical experimental timescale with practical applications such as fed-batch culture in metabolic engineering to see if this approach can be applied.

This is now included in the Discussion when highlighting the main limitations of the optimizer. Since the closed loop system can successfully track changes that occur less rapidly than the oscillator period (Fig. 7bd, Fig. 8d, Supplementary Fig. 17), we expect that the proposed implementation of the optimizer is especially well-suited in contexts where shifts in the culture environment occur slowly (on a timescale ranging from hours to days), which is typical in many fed-batch culture-based metabolic engineering applications.

3. It is not clear whether the extended discussion on Figure 8 should be in the discussion section.

To streamline the Discussion, the extended discussion related to Figure 8 has been integrated into the section focusing on the application examples titled “The optimizer tracks the optimum in diverse contexts.”

Reviewer #2

All of my major concerns were addressed in the revised manuscript. In particular, the study was extended to show biological plausibility of the proposed scheme. The revised version shows examples of biological implementation of the proposed molecular optimizer. Simulations for these optimizers were performed using realistic parameters in literature, and robustness to stochastic noise was also discussed. The reviewer thinks that these extensive computational simulations are sufficient to support the main claim of the paper.

We thank the Reviewer for their constructive feedback and suggestions that helped us not only improve the results presented in the original submission, but also sparked novel lines of research inquiry significantly expanding the scope of our work throughout review process.